# Multiethnic Trends in Early Onset Colorectal Cancer

**DOI:** 10.3390/cancers16020398

**Published:** 2024-01-17

**Authors:** Michelle Nagata, Kohei Miyagi, Brenda Y. Hernandez, Scott K. Kuwada

**Affiliations:** 1Population Sciences in the Pacific Program, University of Hawaii Cancer Center, 701 Ilalo Street, Honolulu, HI 96813, USA; kmiyagi@cc.hawaii.edu (K.M.); brenda@cc.hawaii.edu (B.Y.H.); 2Cancer Biology Program, University of Hawaii Cancer Center, 701 Ilalo Street, Honolulu, HI 96813, USA; skkuwada@hawaii.edu; 3Department of Medicine, John A. Burns School of Medicine, University of Hawaii, 651 Ilalo Street, Honolulu, HI 96813, USA; 4Gastroenterology, The Queen’s Medical Center, 550 South Beretania Street, Honolulu, HI 96813, USA

**Keywords:** colorectal cancer, early onset colorectal cancer, colon cancer, rectal cancer, Hawaii

## Abstract

**Simple Summary:**

Despite overall declines in colorectal cancer (CRC) incidence in the United States and many parts of the world, increases in early onset colorectal cancer (EOCRC) have been observed including disparities by gender and race/ethnicity. While differences in EOCRC have been observed in White, Black, and Hispanic patients, little is known in regard to EOCRC in Asians and Native Hawaiians. The aim of our study was to assess trends in incidence, mortality, and the early onset of CRC in Hawaii’s unique multiethnic population, and identify differences by demographic and clinic-pathologic characteristics. Our results show that significant ethnic disparities in EOCRC exist in Hawaii and suggest the need to tailor standard age-specific CRC screening guidelines in certain ethnicities.

**Abstract:**

Current characteristics of early onset colorectal cancer (EOCRC) in the United States have been mainly studied in Whites, African Americans, and Hispanics, but little is known in regard to EOCRC in Asians and Native Hawaiians in the US. EOCRC was examined in Hawaii’s multiethnic population. Data from the Hawaii Tumor Registry was used to analyze colorectal cancer (CRC) cases diagnosed in Hawaii from 2000–2019 by subsite, age, gender, ethnicity, and stage. Ethnicity analyses were limited to 3524 CRC cases, diagnosed between 2015–2019. Average annual 5-year age-adjusted incidence and mortality rates, average annual percent change over time, and 5-year survival were evaluated. Group comparisons utilized Chi-square and binomial proportion tests. Overall CRC incidence and mortality declined and were more pronounced for colon than rectal/rectosigmoid junction cancers. Colon cancer incidence rates significantly increased 1.46-fold for cases diagnosed under 45 years of age and rectal/rectosigmoid cancers significantly increased 1.54-fold for cases 45–54 years of age. CRC incidence increased sharply for females aged 45–54 years from 2000–2009 to 2010–2019, and increases in colon and rectal/rectosigmoid cancer among individuals aged 45–54 were higher for females. Among both sexes, the increase in rectal/rectosigmoid cancer incidence for individuals under 55 years was highest for stage I cancers. Overall, the mean (SD) age of CRC diagnosis was 5–10 years earlier for Native Hawaiians (60.6 [13.3] years) compared with Japanese, Chinese, Filipinos, Whites, and Other Asians (*p* < 0.001). Native Hawaiians constituted a greater proportion of CRC diagnosed under age 55 years and, conversely, a smaller proportion of cases 55 years and older compared with Japanese, Chinese, Filipinos, Whites, and Other Asians. Native Hawaiians had a significantly higher CRC-related mortality rate (14.5 per 100,000 [95% CI: 12.4, 16.8]) compared with Japanese (10.7 per 100,000 [95% CI: 9.3, 12.3]) and a significantly lower CRC survival rate (62.2% [95% CI: 59.1, 65.2]) compared with Japanese (71.9% [95% CI: 69.9, 73.8]), Filipinos (71.9% [95% CI: 69.2, 74.4]), Chinese (70.2% [95% CI: 65.5, 74.4]), Whites (69.3% [95% CI: 67.1, 71.4]), and Other Asians (71.7% [95% CI: 66.2, 76.5]). In our diverse US population, Native Hawaiians contribute disproportionately to EOCRC and present 5–10 years earlier than Whites, Japanese, Chinese, and Filipinos. EOCRCs are increasing faster in females than males in Hawaii, which differs from trends in the general US population. Emerging ethnic disparities in EOCRC in the US speak to the need for studies on targeted interventions and ethnic-specific risk factors for EOCRC.

## 1. Introduction

Colorectal cancer (CRC) is the third most commonly diagnosed cancer and the second leading cause of cancer-related death worldwide [1,2]. According to the World Health Organization Global Cancer Observatory (GLOBOCAN) database, there were an estimated 1,931,590 new cases of CRC and 935,173 CRC deaths in 2020 [3]. The highest age-standardized incidence and mortality rates for CRC have been observed in Europe (30.0 per 100,000 and 12.6 per 100,000, respectively), Oceania (33.1 and 10.7, respectively), and North America (26.2 and 8.4, respectively) [1,4]. The age-standardized incidence and mortality rates in Japan (38.9 and 12.0, respectively) and Korea (44.5 and 8.7, respectively) reflect regional shifts in CRC trends [4], with higher rectal cancer incidence in Eastern Asia compared with Western countries [5].

Risk factors for CRC include male sex, with incidence rates 30% higher among males than females [2], as well as family history and other genetic and environmental factors [6]. The risk of CRC also increases dramatically with age [4,7]. During 2015–2019 in the United States, incidence rates increased by 80%–100% with each 5-year age group up to 50 years of age and by 20%–30% from ages 55–59 years and older [7]. Changes in guidelines for the initiation of screening in average-risk individuals from age 50 to age 45 in 2018 led to increases in incidence among younger age groups [7].

There has been a steady decline in the incidence of CRC diagnosed in patients 50 years and older, and CRC-related mortality over the past two decades in the US and other high-income countries, which have been primarily attributed to increased CRC screening [2,7]. Early onset CRC (EOCRC), largely defined as CRCs diagnosed before 50 years of age [8,9,10,11,12], is more prominent among males [11,12,13] and is the in the leading cause of cancer-related death in men younger than 50 years in the US [7]. EOCRC incidence is expected to increase for males aged 45–50 [14]. However, US studies have largely focused on Whites, Hispanics, and African Americans [15].

Ethnic disparities in CRC risk have been previously noted. Using national data from 2001–2018, Petrick et al. reported higher CRC rates in American Indians/Alaska Natives (AI/AN) and Blacks compared to Whites [16]. However, increasing EOCRC rates have been reported for Whites as well as AI/AN and Hispanics [16]. Ethnic disparities in CRC risk are suggested to be attributable to differences in both the distribution of known/suspected risk factors and genetic susceptibility between ethnic groups [17].

Ethnic disparities in CRC have been observed in Hawaii, which has a uniquely diverse population predominantly populated by Whites (25.3%), Native Hawaiians (23.6%), Filipinos (17.9%), Japanese (15.0%), Chinese (4.9%), Other Asians (4.0%), and Others (9.3%). In 2015–2019, Hawaii’s overall age-adjusted CRC incidence rate of 39.4 per 100,000 was significantly higher than the US rate of 37.7 per 100,000 [18]. In the Multiethnic Cohort, which enrolled Hawaii and California residents between 1993 and 1996, an increased risk of CRC was found for Japanese men and women, and Black women compared to Whites [17]. Black women and Japanese women also had a higher risk of advanced disease at diagnosis compared to Whites [17]. Black men and women and Japanese women had an increased risk for colon cancer, while Japanese men and women and Native Hawaiian men had an increased risk for rectal cancer compared with Whites [17]. The Multiethnic Cohort reported an increased risk of CRC with excess alcohol consumption among Native Hawaiians, Japanese Americans, Latinos, and Whites and among individuals with a body mass index (BMI) < 25.0, never-users of nonsteroidal anti-inflammatory drugs, and those with lower intakes of dietary fiber and folate [19]. However, no study, to our knowledge, has compared data and statistics on EOCRC in Asian Americans and Native Hawaiians with Whites.

In the present study, we used cancer surveillance data to assess the incidence and mortality rates of the most recent CRC data available for cases diagnosed from 2000 through 2019 in Hawaii by age at diagnosis, gender, ethnicity, cancer stage, and tumor subsite. Understanding ethnic variation in EOCRC is needed to inform targeted interventions and improve prevention.

## 2. Materials and Methods

### 2.1. Study Design

The present analysis utilized de-identified cancer registry data from the Hawaii Tumor Registry (HTR) of the University of Hawaii Cancer Center. The HTR, part of the NCI Surveillance, Epidemiology, and End Results (SEER) Program [20], conducts cancer surveillance for the State of Hawaii and contributes to US estimates of cancer incidence, mortality, and survival as part of SEER. Invasive primary cancers of the colon and rectum diagnosed among Hawaii residents in 2000–2019 were evaluated. Site and histology were coded according to the International Classification of Diseases for Oncology (ICD-O), third edition (ICD-O-3) [21]. Cancer sites and stage at diagnosis were categorized according to SEER definitions [22].

### 2.2. Statistical Analyses

SEER*Stat software 8.4.2 (Informational Management Services, Inc., Calverton, MD, USA) and SAS version 9.4 (SAS Institute, Carey, NC, USA) were used for statistical analyses. Average annual 5-year age-adjusted incidence and mortality rates per 100,000 and 95% confidence intervals (CIs) were calculated. Incidence and mortality rates were limited to categories with at least 10 cases or deaths over the 5-year period. Average annual percent change (AAPC) was calculated using the Joinpoint Regression Program version 5.0.2 (Informational Management Services, Inc., Calverton, MD, USA) [23]. Five-year CRC-specific survival was calculated based on the duration of time from cancer diagnosis to death or last contact.

Comparisons were made by gender, age group, ethnicity, cancer stage, and tumor subsite, which utilized Chi-square tests and binomial proportion tests with significance at *p* < 0.05. Chi-square test with Bonferroni correction was used to analyze proportion differences between ethnic groups with significance at *p* < 0.05. Our study population included Hawaii’s most populous groups (Whites, Japanese, Filipinos, Native Hawaiians, and Chinese) as well as combined groups due to limited numbers (“Other Asians” and “Others”). Individuals with unknown ethnicities were excluded. Racial/ethnic-specific, age-adjusted incidence and mortality rates were calculated only for the most recent five years, 2015–2019, due to the limitation of customized detailed race population data to these years. Hawaii Tumor Registry’s customized detailed race population data covers the most recent five years, 2015–2019, and includes detailed ethnic groups including Japanese, Filipinos, Native Hawaiians, and Chinese [24,25]. Analyses by stage were limited to cases diagnosed between the years of 2005 and 2019 and excluded unstaged cases. Cancer stage was classified according to the TNM staging system.

Comparisons of mean age at diagnosis and 95% CIs were made by t-tests, Wilcoxon tests, Kruskal–Wallis tests, and/or generalized linear models. Pairwise comparisons were performed using the Dwass–Steel–Critchlow–Fligner test.

## 3. Results

### 3.1. Demographic Characteristics

A total of 3524 incident CRC cases were diagnosed in Hawaii from 2015–2019 (Table 1). Males comprised more than half of cases and major racial/ethnic ancestries included Whites, Japanese, Filipinos, and Native Hawaiians (Table 1). Half of cases were stage I or II, and nearly two-thirds were colon tumors.

The mean (standard deviation [SD]) age at diagnosis was 66.0 (14.7) years and was significantly lower for males (64.6 [13.5] years) compared with females (67.7 [15.8] years) (*p* < 0.001) (Table 1). Slightly more than half of CRC cases (53.6%) were 65 years or older at diagnosis. Age at diagnosis significantly differed between ethnic groups ranging from 58.9 (15.0) years for Other ethnic groups to 71.3 (14.5) years for Japanese.

Rectal/rectosigmoid cancers were diagnosed at younger ages (63.0 [13.4] years) compared to colon cancers (67.6 [15.1] years) (*p* < 0.001). A higher proportion of rectal/rectosigmoid cancers were in males (60.7%) compared to females (39.3%) (*p* < 0.001), while the proportions of males and females with colon cancers were not significantly different.

### 3.2. Colorectal Cancer Trends

From 2000–2019, CRC incidence decreased among males from 64.6 per 100,000 (95% CI: 61.8, 67.6) in 2000–2004 to 45.7 per 100,000 (95% CI: 43.6, 47.8) in 2015–2019, representing an average decrease of 2.3% (95% CI: −2.7, −1.9). Over the same period, among females, CRC incidence fell an average of 1.5% (95% CI: −2.2, −0.9) per year from 43.1 per 100,000 (95% CI: 40.9, 45.3) to 33.4 per 100,000 (95% CI: 31.7, 35.2). The declining incidence was more pronounced for colon cancers (average annual decrease of 2.4% [95% CI: −2.9, −1.9] per year) compared to the rectum/rectosigmoid junction (average annual decrease of 0.9% [95% CI: −1.7, −0.2] per year).

Parallel decreases in CRC-related mortality were also observed between 2000 and 2019 (Figure 1). Among males, there was an average decrease of 2.5% (95% CI: −3.8, −1.2) per year, falling from 20.7 per 100,000 (95% CI: 19.1, 22.4) in 2000–2004 to 13.2 per 100,000 (95% CI: 12.1, 14.4) in 2015–2019 (Figure 1). Among females, mortality rates decreased an average of 1.5% (95% CI: −2.8, −0.2) annually falling from 12.4 per 100,000 (95% CI: 11.3, 13.6) in 2000–2004 to 9.7 per 100,000 (95% CI: 8.8, 10.6) in 2015–2019 (Figure 1).

CRC-specific survival significantly differed between cases aged <55 years and 55 years and older but did not differ by gender (Figure 2). Among cases aged 55 years and older, the survival rate was 66.3% (95% CI: 64.9, 67.8) for males and 68.4% (95% CI: 66.8, 70.0) for females (Figure 2). Among individuals aged <55 years, the survival rate was 73.5% (95% CI: 71.0, 75.8) for males and 74.4% (95% CI: 71.4, 77.1) for females (Figure 2).

While CRC incidence declined among individuals 55 years and older, the incidence increased 1.3% (95% CI: 0.2, 2.9) on average annually among individuals aged 45–54 years (1.6% [95% CI: 0.2, 3.1] for males and 2.5% [95% CI: 0.8, 4.4] for females) (Figure 3a,b). Among females aged 45–54 years, the age-adjusted CRC incidence increased sharply from 37.9 per 100,000 (95% CI: 32.4, 44.0) in 2000–2004 and 38.3 per 100,000 (95% CI: 33.0, 44.1) in 2005–2009 to 53.9 per 100,000 (95% CI: 47.5, 60.9) in 2010–2014 and remained elevated through 2015–2019 (52.2 per 100,000 [95% CI: 45.7, 59.4]) (Figure 3c).

For colon cancer specifically, increasing overall incidence was observed among individuals aged <45 years, which rose from 2.6 per 100,000 (95% CI: 2.1, 3.1) in 2000–2004 to 3.8 per 100,000 (95% CI: 3.2, 4.4) in 2015–2019 (Figure 4a). Overall, colon cancer incidence rates were higher in males than females (Figure 4b,c). However, for colon cancers diagnosed under 45 years of age, the incidence rates for females were higher than rates for males in all years except 2005–2009, though differences were not significant (Figure 4a). Among colon cancers diagnosed at 45–54 years, incidence rates significantly increased for females (average annual increase of 1.8% [95% CI: 0.1, 3.6]) from 23.3 per 100,000 (95% CI: 19.1, 28.2) in 2000–2004 to 33.6 per 100,000 (95% CI: 28.6, 39.2) in 2010–2014 (Figure 4b).

Among individuals younger than 45, rectal/rectosigmoid junction cancer incidence increased for females from 1.8 per 100,000 (95% CI: 1.2, 2.5) in 2000–2004 to 2.4 per 100,000 (95% CI: 1.8, 3.2) in 2015–2019 but decreased for males (Figure 5a). Rectal/rectosigmoid cancers increased among individuals aged 45–54 years from 20.3 per 100,000 (95% CI: 17.5, 23.5) in 2000–2004 to 31.3 per 100,000 (95% CI: 27.7, 35.2) in 2015–2019. This increase was higher for females, who had an average annual increase of 3.7% (95% CI: 1.0, 6.9) from 14.6 per 100,000 (95% CI: 11.3, 18.5) in 2000–2004 to 24.1 per 100,000 (95% CI: 19.7, 29.1) in 2015–2019, compared with males whose rectal/rectosigmoid cancer rates increased 2.1% (95% CI: 0.5, 3.7) from 26.0 per 100,000 (95% CI: 21.5, 31.2) in 2000–2004 to 38.6 per 100,000 (95% CI: 33.0, 44.8) in 2015–2019) (Figure 5b). When evaluated by stage at diagnosis for the years 2005–2019, for individuals aged <55 years, rectal/rectosigmoid junction cancer incidence increased significantly only for stage I cancers, which rose 5.4% (95% CI: 2.2, 8.5) (Figure 5c,d).

### 3.3. Ethnic Variation in Colorectal Cancer

In 2015–2019, overall CRC incidence did not significantly differ by ethnicity: 42.6 per 100,000 (95% CI: 36.0, 49.8) for individuals of Other ethnicities, 40.5 per 100,000 (95% CI: 37.5, 43.9) for Japanese, 40.0 per 100,000 (95% CI: 33.8, 47.0) for Other Asians, 39.6 per 100,000 (95% CI: 36.3, 43.2) for Native Hawaiians, 39.4 per 100,000 (95% CI: 36.7, 42.2) for Whites, 37.7 per 100,000 (95% CI: 32.4, 43.7) for Chinese, and 36.8 per 100,000 (95% CI: 33.8, 40.0) for Filipinos.

CRC mortality rates were significantly higher for Native Hawaiians (14.5 per 100,000 [95% CI: 12.4, 16.8]) compared with Japanese (10.7 per 100,000 [95% CI: 9.3, 12.3]) (Figure 6a). CRC mortality rates for the other ethnic groups were as follows: 11.0 per 100,000 (95% CI: 9.4, 12.8) for Filipinos, 11.0 per 100,000 (95% CI: 9.6, 12.5) for Whites, 10.1 per 100,000 (95% CI: 7.7, 13.2) for Chinese, 9.6 per 100,000 (95% CI: 6.8, 13.4) for Other Asians, and 9.2 per 100,000 (95% CI: 6.5, 12.7) for individuals of Other ethnicities (Figure 6a). CRC survival was also significantly lower for Native Hawaiians (63.2% [95% CI: 60.5, 65.8]) compared with Japanese (71.1% [95% CI: 69.5, 72.6]), Filipinos (69.4% [95% CI: 66.9, 71.7]), Whites (68.0% [95% CI: 66.1, 69.8]), and Other Asians (71.2% [95% CI: 66.4, 75.5]) (Figure 6b). The survival rate for Chinese (68.8% [95% CI: 64.9, 72.3]) and individuals of Other ethnicities (65.1% [95% CI: 59.4, 70.2]) was not significantly different compared with the other ethnic groups (Figure 6b).

Age at diagnosis significantly differed between individual ethnic groups (Table 1). The mean age at CRC diagnosis was significantly lower for Native Hawaiians (60.6 [SD = 13.3] years) compared to Japanese (71.3 [SD = 14.5] years, *p* < 0.001), Chinese (69.5 [SD = 15.4] years, *p* < 0.001), Filipinos (65.0 [SD = 14.0] years, *p* < 0.001), Whites (65.6 [SD = 14.0] years, *p* < 0.001), and Other Asians (65.3 [SD = 13.2] years, *p* = 0.004). Age at diagnosis was also significantly lower for Others (58.9 [SD = 15.0] years) compared to Japanese (71.3 [SD = 14.5] years, *p* < 0.001), Chinese (69.5 [SD = 15.4] years, *p* < 0.001), Filipinos (65.0 [SD = 14.0] years, *p* < 0.001), Whites (65.6 [SD = 14.0] years, *p* < 0.001), and Other Asians (65.3 [SD = 13.2] years, *p* = 0.003). Additionally, Japanese were diagnosed at older ages compared to Filipinos (*p* < 0.001), Whites (*p* < 0.001), Other Asians (*p* < 0.001), and Others (*p* < 0.001). Chinese were diagnosed at older ages compared to Filipinos (*p* = 0.011) and Whites (*p* = 0.03). Other Asians had a significantly older age at diagnosis than Others (*p* = 0.003).

Ethnic differences in age at diagnosis were also observed by anatomical subsite. For colon cancer, the mean age at diagnosis was younger for Native Hawaiians (61.7 years [SD = 14.0) compared with Japanese (73.2 years [SD = 14.5], *p* < 0.001), Chinese (72.9 years [SD = 15.0], *p* < 0.001), Whites (66.8 years [SD = 14.7], *p* < 0.001), and Filipinos (66.6 years [SD = 13.9], *p* < 0.001) (Figure 7a). The mean age at colon cancer diagnosis was also younger for individuals of Other ethnicities (60.2 years [SD = 15.6]) compared with Japanese (*p* < 0.001), Chinese (*p* < 0.001), Whites (*p* < 0.001), and Filipinos (*p* = 0.002) (Figure 7a). For rectal/rectosigmoid cancer, the mean age at diagnosis was younger for Native Hawaiians (58.6 years [SD = 11.6]) compared with Japanese (67.3 years [SD = 13.7], *p* < 0.001), Other Asians (64.7 years [SD = 13.1], *p* = 0.012), Whites (63.3 years [SD = 12.3], *p* < 0.001), Filipinos (62.6 years [SD = 13.8], *p* = 0.014) (Figure 7b). The mean age at rectal/rectosigmoid cancer diagnosis was also younger for Others (56.4 years [SD = 13.5]) compared with Japanese (*p* < 0.001), Other Asians (*p* = 0.012), Whites (*p* = 0.003), and Filipinos (*p* = 0.024) (Figure 7b).

Ethnic group variations were observed in the proportions of cases contributing to CRC diagnoses under the age of 55 years compared with diagnoses at 55 years and older (Figure 8). Native Hawaiians had a significantly higher proportion of CRCs diagnosed under the age of 55 years and a significantly lower proportion diagnosed at 55 years and older compared with Japanese (*p* < 0.001), Chinese (*p* < 0.001), Filipinos (*p* < 0.001), Whites (*p* < 0.001), and Other Asians (*p* < 0.001) (Figure 8). The proportion of CRCs diagnosed under the age of 55 years in Native Hawaiians (34.5%) was over two times the proportions observed in Japanese (14.2%) and Chinese (16.3%) younger than 55 years (Figure 8).

Japanese had a significantly higher proportion of CRCs diagnosed at the age of 55 and older and a significantly lower proportion diagnosed under the age of 55 compared with Native Hawaiians, Filipinos (*p* < 0.001), Whites (*p* < 0.001), and Others (*p* < 0.001) (Figure 8). Notably, in 2015–2019, colon cancers increased 4.1% (95% CI: 0.8, 7.8) per year among Japanese aged 75 years and older.

While Chinese and Whites each had a significantly higher proportion of CRCs diagnosed at the age of 55 and older and a significantly lower proportion diagnosed under the age of 55 compared with Others (*p* < 0.001), differences with the other ethnic groups were not significant (Figure 8). Filipinos and Other Asians had similar proportions of CRCs diagnosed under 55 years of age and age 55 and older.

## 4. Discussion

This study evaluated CRC incidence, mortality, and survival in Hawaii from 2000 through 2019. Overall, our findings corroborate observed increases in EOCRC incidence in the US despite overall declines in CRC [7,14]. In our study, the declining CRC incidence and mortality from 2000–2019 was more pronounced for the colon than the rectum/rectosigmoid junction. Consistent with other studies [13,14,26], we observed a younger age at diagnosis for rectal/rectosigmoid cancers compared to colon cancers. A recent estimate projected a 31% increase in rectal cancer incidence for US individuals aged 45–50 years from 2018 through 2030 in another study [14]. In our statewide population, increasing CRC incidence was substantially greater for stage I cancers among individuals aged <55 years, which, in part, may be attributed to recent changes in screening guidelines or more endoscopic evaluation of younger individuals with lower gastrointestinal symptoms, especially rectal bleeding.

In our study, males had higher 5-year CRC incidence rates than females across all age groups as well as a younger age at CRC diagnosis. However, CRC incidence increased sharply for females aged 45–54 years from 2000–2009 to 2010–2019. In our ethnically diverse population, increases in rectal/rectosigmoid cancer incidence rates among individuals aged 45–54 were higher for females. This contrasts with national reports of higher CRC incidence [2,7] and higher EOCRC incidence for males compared with females [13,14], though CRC cases younger than 50 years in the US were more likely to be women compared to cases aged 50 and older [7]. Women have been reported to have higher rates of early onset distal colon cancer [16], which, along with rectal tumors, are typically associated with younger age groups [26]. For women, the largest proportion of EOCRCs were rectal, whereas the largest proportion of later-onset CRCs were proximal colon cancers [16]. Our study found that rectal/rectosigmoid cancer incidence among cases under 45 years of age declined for males from 2000–2019 but increased steadily for females. We also observed higher increases in colon and rectal/rectosigmoid junction cancers for women than men aged 45–54 years. There is limited data to propose possible risk factors that may be responsible for these differences by gender. While the potential role for human papillomavirus (HPV) in rectal adenocarcinomas remains controversial [27,28,29], there were no increased rates of HPV-related cervical cancers in Hawaii during this study time period [30]. Higher body mass index (BMI) has been correlated with CRC risk [31] and appears to be greater among females [32]. The average BMI in Hawaii has increased dramatically over the past few decades [33].

While most studies in the US investigated EOCRC in White, African American, and Hispanic/Latino populations [15], others have studied Asian Americans and Pacific Islanders (AAPI) in aggregate [16,34,35]. Our study was unique in that it compared Whites, Japanese, Filipinos, Native Hawaiians, and Chinese living in the US. Native Hawaiians and those of Other ethnicities (mostly other Pacific Islanders) had a significantly earlier mean age at colon and rectal/rectosigmoid cancer diagnosis compared to all other ethnic groups, with the greatest age differences relative to Japanese. Native Hawaiians also had significantly higher CRC mortality compared with Japanese and significantly lower CRC survival compared with Japanese, Chinese, Filipinos, Whites, and Other Asians. In contrast to previous studies that assessed Asian Americans and Pacific Islanders (AAPI) as one racial group [16] or combined AI/AN and AAPI into one group due to low frequency [14], the observed disparities in age at diagnosis and proportion of EOCRC support disaggregation of Asians and Pacific Islanders in studies on EOCRC.

We found significant differences in the proportions of CRCs diagnosed at ages younger than 55 years and 55 years and older across ethnic groups. Native Hawaiians contributed a significantly higher proportion of CRCs diagnosed at ages younger than 55 and a significantly lower proportion of CRCs diagnosed at 55 years and older compared with Japanese, Chinese, Filipinos, Whites, and Other Asians. This pattern was also seen with age at diagnosis, where the mean age of CRC diagnosis was over a decade younger for Native Hawaiians compared with Japanese. These findings are similar to what has been observed in African Americans and Hispanics in the Southern U.S. who contribute disproportionately to EOCRC compared with Whites [15,26,36]. Interestingly, CRC characteristics among Japanese, Chinese, Filipinos, and Whites in Hawaii were more similar than those observed in Native Hawaiians and those of Other ethnicities. The reasons for these ethnic disparities in Hawaii are presently unknown, but the high prevalence of obesity among Native Hawaiians is proposed to contribute to increased CRC risk and poor CRC survival [37].

The limitations of our study included a lack of data on risk factors for CRC such as social determinants of health, access to healthcare, CRC screening history, family history, hereditary cancer syndromes, and chronic inflammatory colitis. We did not find significant differences in CRC incidence by ethnicity, possibly due to the limited case numbers of disaggregated AAPI ethnic groups. Moreover, ethnicity analyses were also limited to 2015–2019 when customized detailed race population data was available.

EOCRC patients tend to have a longer duration of symptoms at presentation and delayed time to diagnosis compared to older patients [2,7]. National trends also reflect a rise in advanced CRCs after the initiation of widespread screening [7]. While diagnosis at the advanced stages does not necessarily confer increased risk of death, as EOCRC patients have shown higher disease-specific survival at every stage despite more adverse histologic features [38,39], this may be due to fewer comorbidities and more aggressive treatment [40,41]. Thus, it remains critical to investigate the factors contributing to increasing EOCRC incidence rates.

The etiology of sporadic EOCRC remains unclear. Risk factors for later-onset CRC, such as a westernized diet and overweight and obesity have been proposed to contribute to the rise in EOCRC in highly westernized countries including Australia, Canada, New Zealand, the United Kingdom, and the United States [42]. Type 2 diabetes (T2D), which shares these CRC risk factors, has been implicated as a risk factor for CRC [15,43]. Gut microbiome alterations, which may be induced by diet, exercise, and weight change, have also been proposed to contribute to EOCRC through inflammatory pathways [8,42]. Still, there is inconsistent data supporting an etiological role for gut microbiome dysbiosis in CRC development [4]. Assessments of socioeconomic risk factors for CRC have indicated no significant differences in household income and education level between patients with EOCRC, patients with later-onset CRC, and controls [13], but this has not been studied in Hawaii. Hereditary cancer syndromes, such as Lynch syndrome, were responsible for approximately 22% of EOCRC cases in one study [8]. However, over half of EOCRC patients lack family history of CRC or hereditary cancer syndromes [8].

## 5. Conclusions

In contrast with Hawaii’s overall declining CRC incidence, an increasing incidence was observed for colon cancer among individuals younger than 45 years of age, rectal/rectosigmoid junction cancer among individuals aged 45–54, and stage I colon cancer under 55 years. The rising incidence of colon and rectal/rectosigmoid junction cancers in individuals aged 45–54 was more pronounced for females, which differs from trends in the general US population. Notable disparities were observed for Native Hawaiians including earlier onset, higher mortality, and poorer survival compared to other ethnicities. Emerging ethnic disparities in EOCRC in the US speak to the need for studies on targeted interventions and ethnic-specific risk factors for EOCRC. Further investigation of risk factors for EOCRC may inform preventive and diagnostic interventions in understudied ethnic groups such as Native Hawaiians, as well as African Americans and Hispanics, in whom EOCRC is overrepresented in the US.

## Figures and Tables

**Figure 1 cancers-16-00398-f001:**
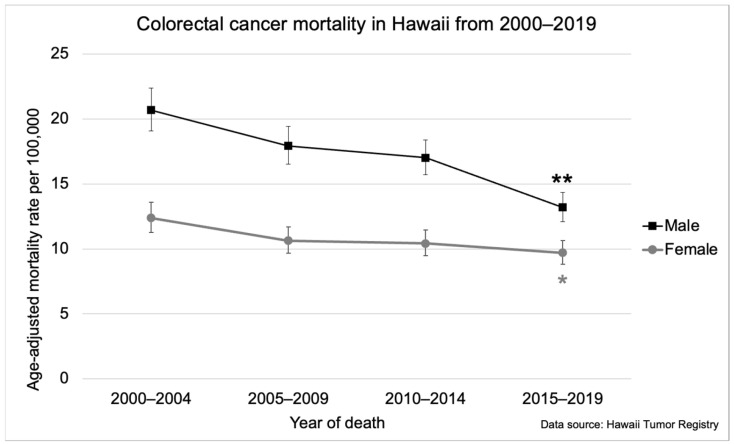
Colorectal cancer-related mortality in Hawaii from 2000–2019 by gender. The asterisk (*) indicates a significant difference from the 2000–2004 incidence rate. The double asterisk (**) indicates a significant difference from the 2000–2004, 2005–2009, and 2010–2014 incidence rates. Five-year mortality rates were age-adjusted to the 2000 US Standard Population.

**Figure 2 cancers-16-00398-f002:**
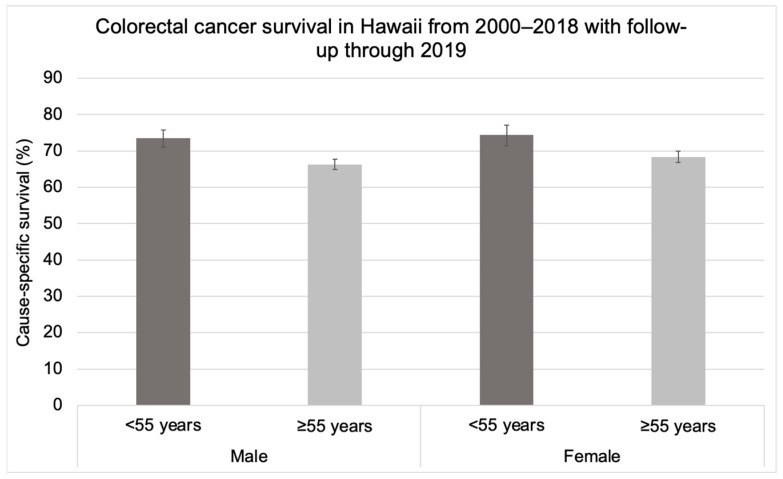
Colorectal cancer-specific survival in Hawaii from 2000–2018 with follow-up through 2019 by gender among individuals aged 55 years and older at diagnosis and younger than 55 years at diagnosis.

**Figure 3 cancers-16-00398-f003:**
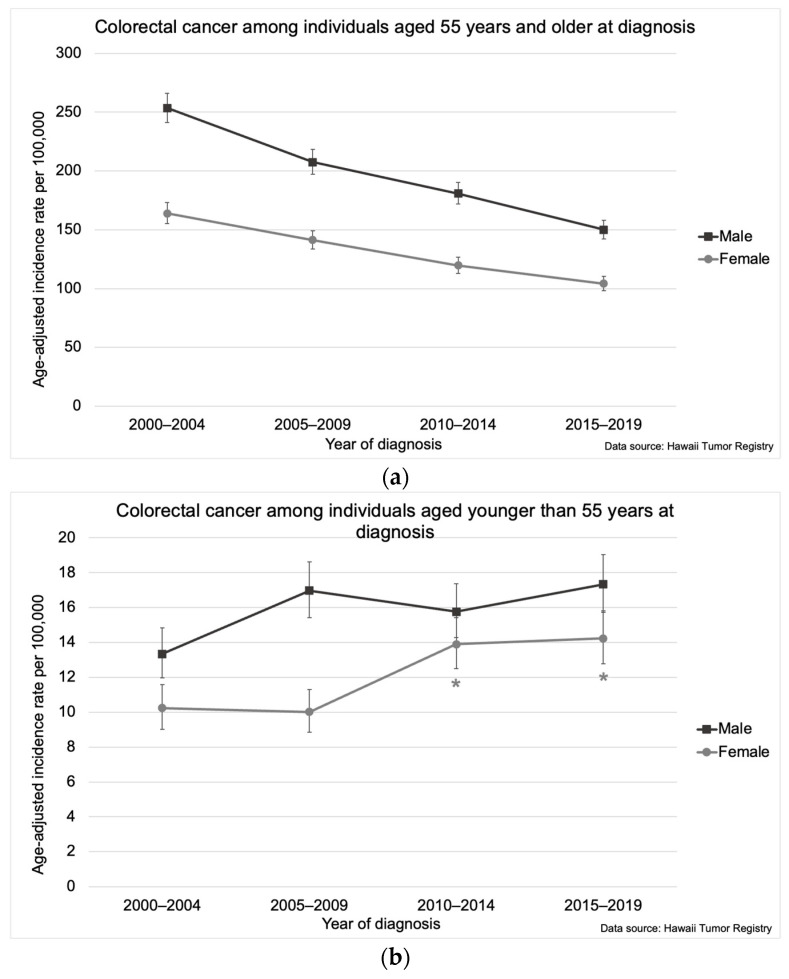
Colorectal cancer incidence in Hawaii from 2000–2019 by gender among individuals aged (**a**) 55 years and older at diagnosis, (**b**) younger than 55 years at diagnosis, and (**c**) 45–54 years at diagnosis. The asterisk (*) indicates a significant difference from 2000–2004 and 2005–2009 incidence rates. Five-year incidence rates per 100,000 were age-adjusted to the 2000 US Standard Population.

**Figure 4 cancers-16-00398-f004:**
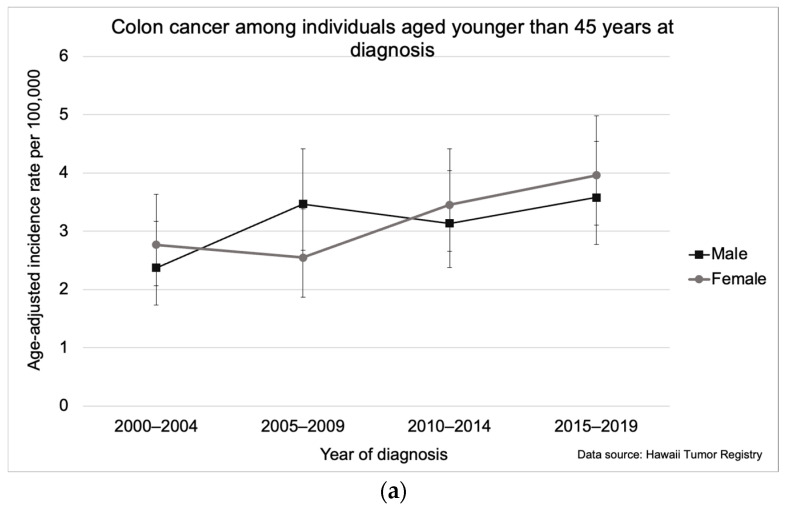
Colon cancer incidence in Hawaii from 2000–2019 by gender for individuals aged (**a**) younger than 45 years, (**b**) 45–54, (**c**) 55–64, and 65 years and older. The asterisk (*) indicates a significant difference from the 2000–2004 incidence rate. Five-year incidence rates per 100,000 were age-adjusted to the 2000 US Standard Population.

**Figure 5 cancers-16-00398-f005:**
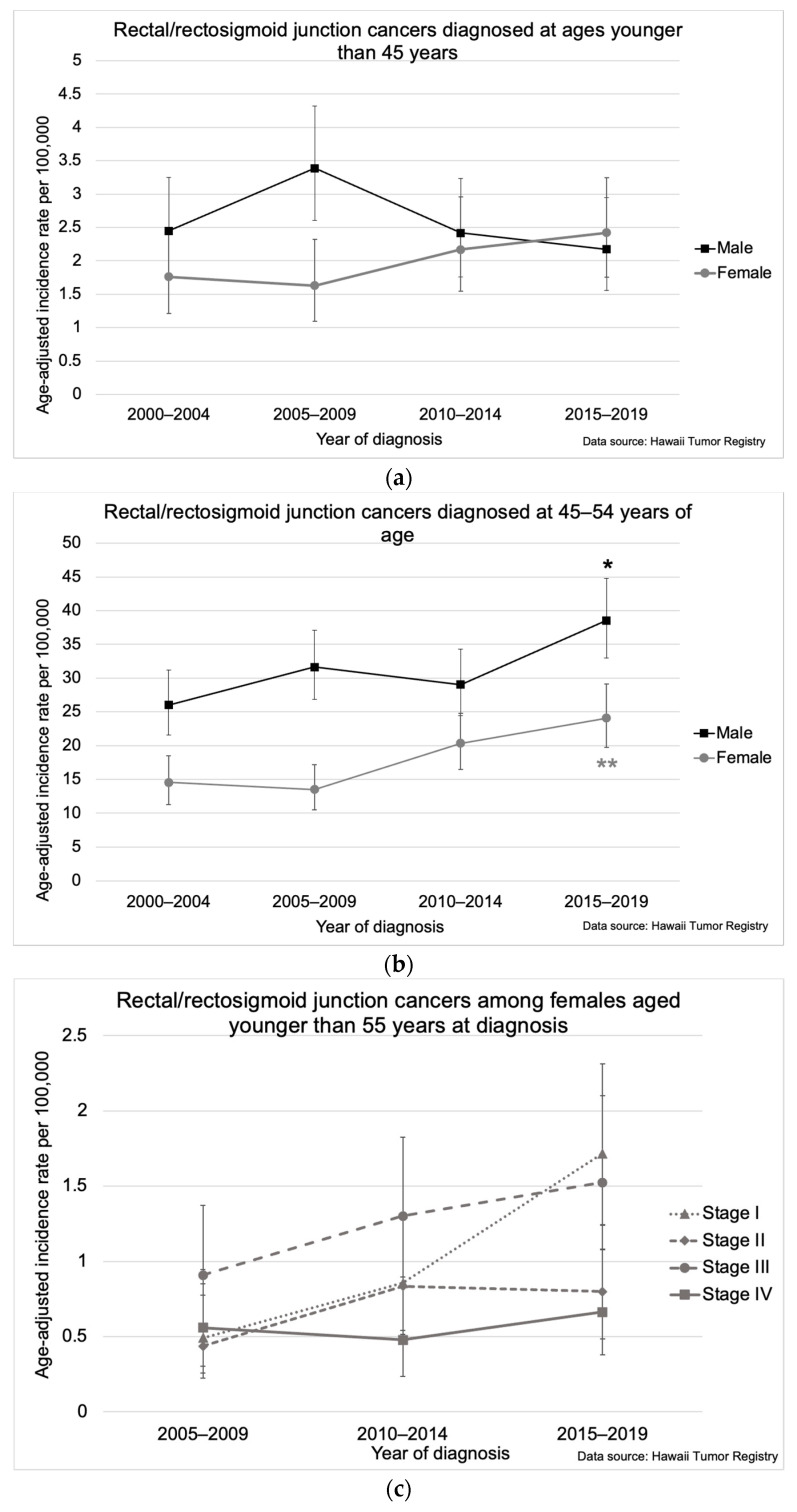
Rectal/rectosigmoid junction cancer incidence in Hawaii from 2000–2019 (**a**) by age at diagnosis for males and females aged younger than 45 years, (**b**) by age at diagnosis for males and females aged 45–54 years, (**c**) by stage at diagnosis for females aged younger than 55 years, and (**d**) by stage at diagnosis for males aged younger than 55 years. The asterisk (*) indicates a significant difference from the 2000–2004 incidence rate. The double asterisk (**) indicates a significant difference from 2000–2004 and 2005–2009 incidence rates. Five-year incidence rates per 100,000 were age-adjusted to the 2000 US Standard Population. Incidence rates for stages were limited to the years 2005–2019.

**Figure 6 cancers-16-00398-f006:**
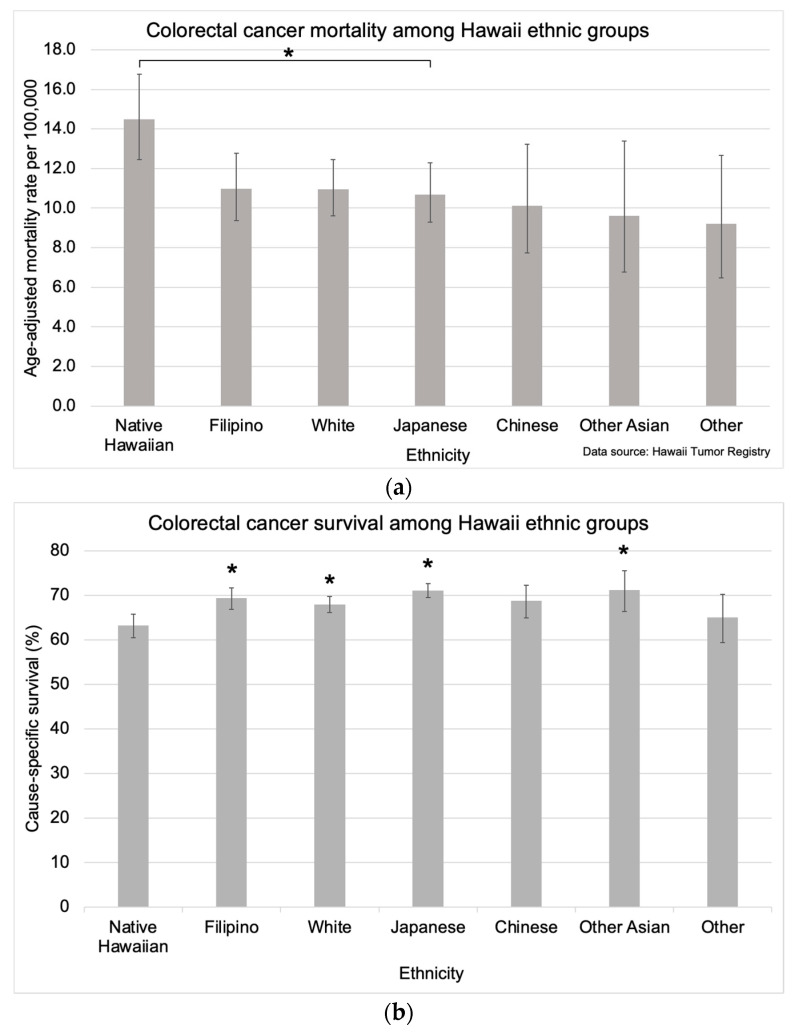
Colorectal cancer-specific (**a**) mortality and (**b**) survival in Hawaii by ethnicity. Mortality calculations were limited to data available from 2015–2019, and data for survival calculations was available from 2000–2018 with follow-up through 2019. The asterisk (*) indicates a significant difference from the mortality rate of Native Hawaiians. Five-year mortality rates were age-adjusted to the 2000 US Standard Population.

**Figure 7 cancers-16-00398-f007:**
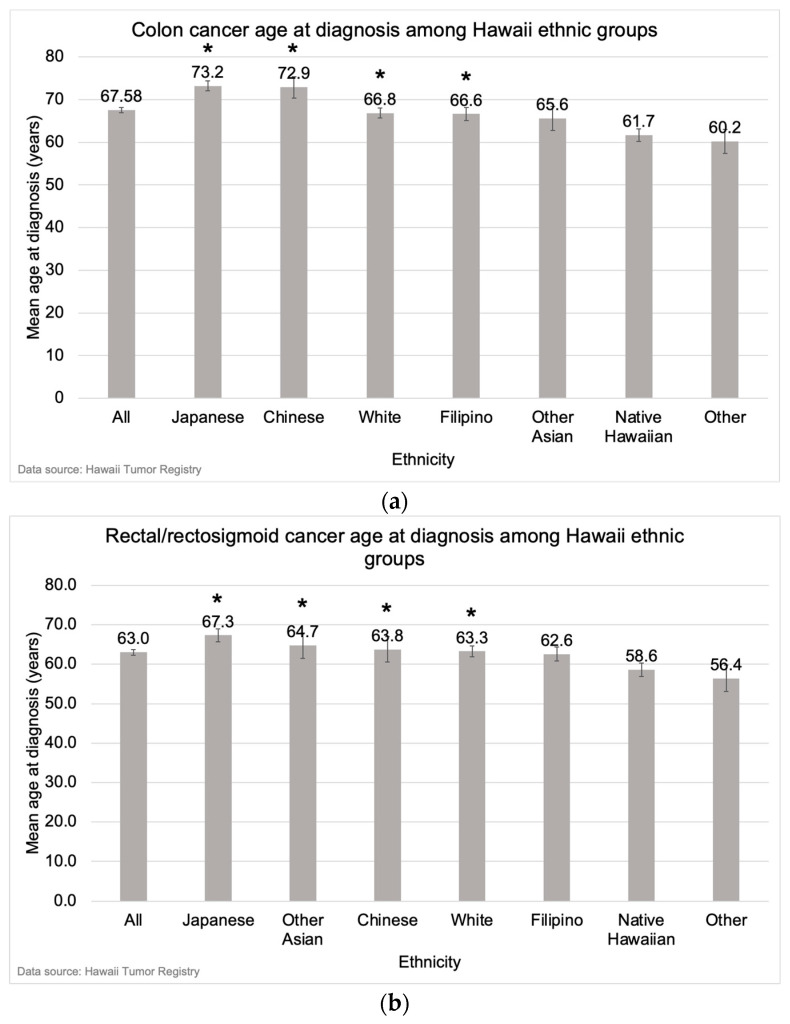
Mean age at diagnosis for (**a**) colon cancer and (**b**) rectal/rectosigmoid junction cancer in Hawaii from 2015–2019 by ethnicity. The asterisk (*) indicates a significant difference from the mean age at diagnosis of Native Hawaiians and individuals of Other ethnicities.

**Figure 8 cancers-16-00398-f008:**
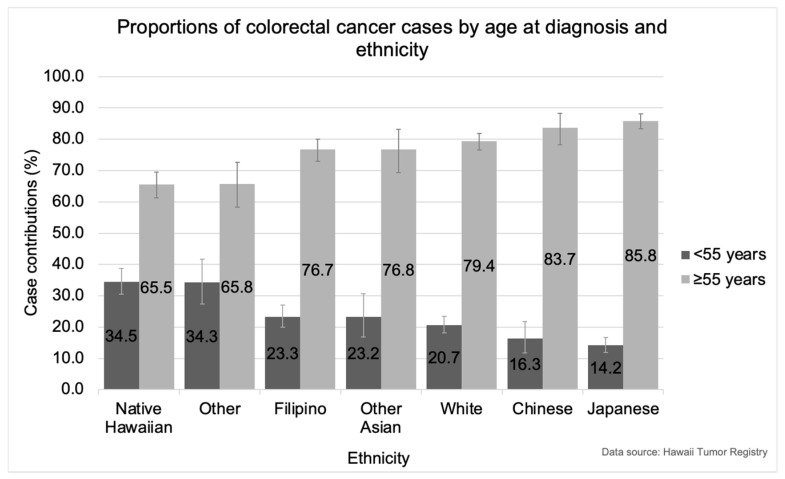
Proportions of colorectal cancer cases diagnosed in Hawaii from 2015–2019 by age at diagnosis and ethnicity, both genders combined. Comparisons between individual ethnic groups that are significant at the level of Bonferroni correction applied alpha = 0.0024 include: Native Hawaiian and Japanese; Native Hawaiian and Chinese; Native Hawaiian and Filipino; Native Hawaiian and White; Native Hawaiian and Other Asian; Japanese and Filipino; Japanese and White; Japanese and Other; Chinese and Other; and White and Other.

**Table 1 cancers-16-00398-t001:** Demographic characteristics of colorectal cancer cases diagnosed in Hawaii from 2015–2019.

Demographic Characteristic	No. (%) ^a^ *n* = 3524	Mean (SD) Age at Diagnosis, Years	*p*-Value ^b^
Age at diagnosis		66.0 (14.7)	–
<45	228 (6.5)		
45–54	560 (15.9)		
55–64	846 (24.0)		
≥65	1890 (53.6)		
Gender			<0.001
Male	1932 (54.8)	64.6 (13.5)	
Female	1592 (45.2)	67.7 (15.8)	
Ethnicity			<0.001
White	959 (27.5)	65.6 (14.0)	
Japanese	873 (25.0)	71.3 (14.5)	
Filipino	570 (16.3)	65.0 (14.0)	
Native Hawaiian	534 (15.3)	60.6 (13.3)	
Chinese	221 (6.3)	69.5 (15.4)	
Other Asian	155 (4.4)	65.3 (13.2)	
Other	181 (5.2)	58.9 (15.0)	
Cancer stage ^c^			<0.001
I	779 (26.3)	63.6 (14.4)	
II	705 (23.8)	68.9 (13.8)	
III	847 (28.6)	65.5 (14.3)	
IV	629 (21.3)	65.3 (14.2)	
Tumor subsite			<0.001
Colon	2271 (64.4)	67.6 (15.1)	
Rectum/rectosigmoid junction	1253 (35.6)	63.0 (13.4)	

^a^ Counts and percentages exclude cases with unknown ethnicity and/or stage. ^b^ *p*-values for comparisons of mean age at diagnosis across groups were calculated by two-sample Wilcoxon tests and Kruskal–Wallis tests for three or more groups. ^c^ Counts for cancer stage were limited to cases diagnosed between 2005–2019 and excluded unstaged cases.

## Data Availability

The data presented in this study are available on request from the corresponding author. The data are not publicly available due to privacy restrictions.

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
