# Peer review of "Multiethnic Trends in Early Onset Colorectal Cancer"

_cancers, 2024, doi:10.3390/cancers16020398_

Round 1
Reviewer 1 Report
Comments and Suggestions for Authors
Congratulations for your present work. It is an interesting analysis of trends in cancer incidence, mortality and survival in Hawaii. Introduction is comprehensive enough and Material and Methods are properly described and are appropriate for the objectives of the study. The results are presented fairly clearly, but may need some restructuring. Discussion/Conclusions are well substantiated with the results.
As a minor revision, the following should be taken into account:
- Materials and Methods: Line 105. SEER is first mentioned at this point. It should be cited in full as ‘Surveillance, Epidemiology, and End Results Program’.
- Materials and Methods: Line 117. Is it relative or observed survival?. You need to clarify this.
- Results: Legend table 1. Perhaps it should be added in section ‘c’ that unstaged cases were excluded.
- Results: To ensure an easy reading of Results you may consider repeating the same structure for incidence figures for colorectal, colon and rectal/rectosigmoid junction cancers if you are able to reanalyze data. I.e. <45 years, 45-54 years, 55+ years, and stage distribution for male and females (<55 and 55+). Probably, it would be more informative for the reader.
- Results: Mortality trend data need a Figure in Results as well as Survival data.
- Results. Ethnic Variation in Colorectal Cancer: Mortality data need a Figure in Results or Supplementary Material as well as Survival.
- Results. Ethnic Variation in Colorectal Cancer: Table 2 is redundant with Figure 5. You can consider to remove it.
- Discussion: Line 321. AAPI is first mentioned at this point. It should be cited in full as ‘Asian Americans and Pacific Islanders’.
Best regards
Author Response
Thank you for taking the time to review this manuscript. Please find the detailed responses below and the corresponding revisions in track changes in the resubmitted file.
Comment 1: Materials and Methods: Line 105. SEER is first mentioned at this point. It should be cited in full as ‘Surveillance, Epidemiology, and End Results Program’.
Response 1: Thank you for your suggestion. We revised the first mention of SEER to be cited in full.
Comment 2: Materials and Methods: Line 117. Is it relative or observed survival? You need to clarify this.
Response 2: Thank you for your question. We revised line 119 to clarify that five-year survival was relative according to SEER guidelines.
Comment 3: Results: Legend table 1. Perhaps it should be added in section ‘c’ that unstaged cases were excluded.
Response 3: We agree and added that unstaged cases were excluded in the Table 1 legend on line 150.
Comment 4: Results: To ensure an easy reading of Results you may consider repeating the same structure for incidence figures for colorectal, colon and rectal/rectosigmoid junction cancers if you are able to reanalyze data. I.e. <45 years, 45-54 years, 55+ years, and stage distribution for male and females (<55 and 55+). Probably, it would be more informative for the reader.
Response 4: Thank you for your suggestion. We have reanalyzed data according to these age groups. We revised our updated Figure 3 (original Figure 1) to show CRC incidence among the < 55 and ≥ 55 age groups and significant changes in CRC incidence among females in the 45–54 age group. The stage distribution comparisons among rectal/rectosigmoid junction cancer cases in our updated Figure 5c–d were also changed to cases < 55 years.
Comment 5: Results: Mortality trend data need a Figure in Results as well as Survival data.
Response 5: We added figures for overall CRC-related mortality (Figure 1) and survival (Figure 2) as well as text describing survival differences between < 55 and ≥ 55 age groups on lines 184– 188.
Comment 6: Results. Ethnic Variation in Colorectal Cancer: Mortality data need a Figure in Results or Supplementary Material as well as Survival.
Response 6: We agree and added Figure 6 for CRC-related mortality and survival by ethnicity.
Comment 7: Results. Ethnic Variation in Colorectal Cancer: Table 2 is redundant with Figure 5. You can consider to remove it.
Response 7: Thank you for your suggestion. We agree and removed Table 2. In the updated Figure 8 (original Figure 5) legend, we noted the significant comparisons that were in Table 2.
Comment 8: Discussion: Line 321. AAPI is first mentioned at this point. It should be cited in full as ‘Asian Americans and Pacific Islanders’.
Response 8: We revised the first mention of AAPI to “Asian Americans and Pacific Islanders.”
Reviewer 2 Report
Comments and Suggestions for Authors
The authors aimed to assess trends in incidence, mortality, and early onset of CRC in Hawaii’s unique multiethnic population and identify differences by demographic and clinic- pathologic characteristics. They finally showed that significant ethnic disparities in EOCRC exist in Hawaii and suggest the need to tailor standard age-specific CRC screening guidelines in certain ethnicities. It is meaningful and interesting. There are several points which need to be addressed.
1. How about the reason of analyzing colorectal cancer (CRC) cases diagnosed in Hawaii from 2000–2019?
2. Figure 1, Figure 2, Figure 3, Figure 4, Figure 5 can be revised such as using graphpad software.
3. The statistical differences can be noted with “*” or other symbols to more clearly describe the differences between groups.
4. How about the authors’ opionion on the results “Early-onset colon cancers is increasing faster in females than males in Hawaii, which differs from trends in the general US population.”?
Comments on the Quality of English Language
Average
Author Response
Thank you for taking the time to review this manuscript. Please find the detailed responses below and the corresponding revisions in track changes in the resubmitted file.
Comment 1: How about the reason of analyzing colorectal cancer (CRC) cases diagnosed in Hawaii from 2000–2019?
Response 1: Thank you for your question. We analyzed the most recent CRC data available for cases diagnosed in Hawaii from 2000–2019. The materials and methods section has been updated this on line 98. We focused specifically on CRC cases diagnosed in Hawaii due to Hawaii’s uniquely diverse population predominantly populated by Whites, Native Hawaiians, Filipinos, Japanese, Chinese, other Asians, and others (mostly Pacific Islanders) living in the US. This is described in the introduction in lines 81–85. Ethnic disparities in CRC are also noted in the introduction. Most studies in the US investigated EOCRC in White, African American, and Hispanic/Latino populations as well as Asian Americans and Pacific Islanders in aggregate. Our study identified significant differences in CRC incidence between disaggregated AAPI ethnic groups.
Comment 2: Figure 1, Figure 2, Figure 3, Figure 4, Figure 5 can be revised such as using GraphPad software.
Response 2: We have revised each figure to show asterisks indicating significant differences and consistent age group categories (< 45, 45–54, 55–64, and ≥ 65 years; < 55 and ≥ 55 years for stage distribution) throughout all analyses.
Comment 3: The statistical differences can be noted with “*” or other symbols to more clearly describe the differences between groups.
Response 3: We have added asterisks (*) to indicate significant differences in our figures.
Comment 4: How about the authors’ opinion on the results “Early-onset colon cancers is increasing faster in females than males in Hawaii, which differs from trends in the general US population.”?
Response 4: Thank you for your question. We elaborate upon our observation of faster increases in EOCRC for females compared to males in Hawaii in the discussion section from lines 498– 517. There is limited data to explain this difference in Hawaii. Human papillomavirus (HPV) infection has been proposed as a possible risk factor in rectal adenocarcinomas, but the association remains controversial. Moreover, rates of HPV-related cervical cancers were not reported to have increased in Hawaii during this study time period. Higher body mass index (BMI) has been correlated with CRC risk. The average BMI has increased in Hawaii over the
past few decades and appears to be greater among females. However, further research is needed to explain faster increases in EOCRC among females in Hawaii.
Round 2
Reviewer 2 Report
Comments and Suggestions for Authors
The figures can be revised with high-solution versions.
Comments on the Quality of English Languageaverage
Author Response
Thank you for taking the time to review this manuscript. Please find the detailed responses below and the corresponding revisions in track changes in the resubmitted file.
Comment 1: The figures can be revised with high-solution versions.
Response 1: Thank you for your suggestion. We reached out to the editors of Cancers for guidance on our figures. The editors replied that the resolution of the figures is acceptable according to the journal’s requirements. We increased the font size in each figure and uploaded a zip file containing all figures in JPEG format.
